Stimulation of glucose uptake in murine soleus muscle and adipocytes by 5-(4-phenoxybutoxy)psoralen (PAP-1) may be mediated by Kv1.5 rather than Kv1.3

Ngala Robert A. 1
Zaibi Mohamed S.
Langlands Kenneth
Stocker Claire J.
Arch Jonathan R.S. jon.arch@buckingham.ac.uk
Cawthorne Michael A.
Clore Laboratory, Buckingham Institute for Translational Medicine, University of Buckingham , Buckingham , UK
Berdeaux Rebecca
1 Current affiliation: Department of Molecular Medicine, School of Medical Sciences, Kwame Nkrumah University of Science and Technology Kumasi, Ghana

Electronic publication date: 2014 Oct 7
Publication date: 2014
Volume: 2
Electronic Location ID: e614
Received 2013 Sep 10; Accepted 2014 Sep 17
Copyright: © 2014 Ngala et al.
Copyright year: 2014
Copyright holder: Ngala et al.
License: This is an open access article distributed under the terms of the Creative Commons Attribution License, which permits unrestricted use, distribution, and reproduction in any medium, provided the original author and source are credited.
License URL: https://creativecommons.org/licenses/by/3.0/

Keywords: Potassium channel, PAP-1, Glucose uptake, Adipocyte, Soleus muscle, TNFα, Kv1.3, Kv1.5

Funding: The University of Buckingham Xention Limited 05/045 This work was funded by The University of Buckingham, supported in part by grant 05/045 from Xention Limited. The funders had no role in study design, data collection and analysis, decision to publish, or preparation of the manuscript.

==============================
Kv1 channels are shaker-related potassium channels that influence insulin sensitivity. Kv1.3−/− mice are protected from diet-induced insulin resistance and some studies suggest that Kv1.3 inhibitors provide similar protection. However, it is unclear whether blockade of Kv1.3 in adipocytes or skeletal muscle increases glucose uptake. There is no evidence that the related channel Kv1.5 has any influence on insulin sensitivity and its expression in adipose tissue has not been reported. PAP-1 is a selective inhibitor of Kv1.3, with 23-fold, 32-fold and 125-fold lower potencies as an inhibitor of Kv1.5, Kv1.1 and Kv1.2 respectively. Soleus muscles from wild-type and genetically obese ob/ob mice were incubated with 2-deoxy[1-14C]-glucose for 45 min and formation of 2-deoxy[1-14C]-glucose-6-phosphate was measured. White adipocytes were incubated with D-[U-14C]-glucose for 1 h. TNFα and Il-6 secretion from white adipose tissue pieces were measured by enzyme-linked-immunoassay. In the absence of insulin, a high concentration (3 µM) of PAP-1 stimulated 2-deoxy[1-14C]-glucose uptake in soleus muscle of wild-type and obese mice by 30% and 40% respectively, and in adipocytes by 20% and 50% respectively. PAP-1 also stimulated glucose uptake by adipocytes at the lower concentration of 1 µM, but at 300 nM, which is still 150-fold higher than its EC50 value for inhibition of the Kv1.3 channel, it had no effect. In the presence of insulin, PAP-1 (3 µM) had a significant effect only in adipocytes from obese mice. PAP-1 (3 µM) reduced the secretion of TNFα by adipose tissue but had no effect on the secretion of IL-6. Expression of Kv1.1, Kv1.2, Kv1.3 and Kv1.5 was determined by RT-PCR. Kv1.3 and Kv1.5 mRNA were detected in liver, gastrocnemius muscle, soleus muscle and white adipose tissue from wild-type and ob/ob mice, except that Kv1.3 could not be detected in gastrocnemius muscle, nor Kv1.5 in liver, of wild-type mice. Expression of both genes was generally higher in liver and muscle of ob/ob mice compared to wild-type mice. Kv1.5 appeared to be expressed more highly than Kv1.3 in soleus muscle, adipose tissue and adipocytes of wild-type mice. Expression of Kv1.2 appeared to be similar to that of Kv1.3 in soleus muscle and adipose tissue, but Kv1.2 was undetectable in adipocytes. Kv1.1 could not be detected in soleus muscle, adipose tissue or adipocytes. We conclude that inhibition of Kv1 channels by PAP-1 stimulates glucose uptake by adipocytes and soleus muscle of wild-type and ob/ob mice, and reduces the secretion of TNFα by adipose tissue. However, these effects are more likely due to inhibition of Kv1.5 than to inhibition of Kv1.3 channels.

Introduction

Kv1 channels are shaker-related, voltage-gated potassium channels (Gutman et al., 2003). It has been proposed that compounds that block the Kv1.3 channel might be used in the treatment of type 2 diabetes and obesity (Choi & Hahn, 2010). Thus, a variant of the human Kv1.3 gene is associated with low insulin sensitivity and impaired glucose tolerance (Tschritter et al., 2006), whereas Kv1.3−/− mice were protected from diet-induced insulin resistance (Xu et al., 2003) and obesity (Xu et al., 2004). Some studies on inhibitors of Kv1.3 have demonstrated their ability to improve insulin sensitivity in mice (Xu et al., 2004; Upadhyay et al., 2013), but another has failed to show any benefit (Straub et al., 2011).

In those studies in which improved insulin sensitivity has been shown, the tissues where the inhibitors have their primary action are uncertain. Kv1.3 channels in liver, postganglionic sympathetic neurons and brown adipose tissue are suggested targets (Upadhyay et al., 2013). It is unclear whether white adipose tissue or skeletal muscle could be the primary site of action because there is disagreement as to whether Kv1.3 is expressed, particularly at the protein level, in these tissues (Grande et al., 2003; Li et al., 2006; Straub et al., 2011; Upadhyay et al., 2013). Nevertheless, the Kv1.3 inhibitor margatoxin decreased TNFα secretion by white adipose tissue from genetically obese (ob/ob) mice (Xu et al., 2004), and administration of the highly selective Kv1.3 inhibitor ShK-186 for 45 days to mice fed on an obesity-inducing diet reduced TNFα mRNA in visceral adipose tissue (Upadhyay et al., 2013). The target for the latter effect might be Kv1.3 channels in inflammatory cells, such as macrophages. Decreased inflammation of adipose tissue, including decreased secretion of TNFα, would be expected to improve insulin sensitivity (Calle & Fernandez, 2012).

As part of an investigation into the potential of Kv1.3 inhibitors for the treatment of type 2 diabetes, we have studied the effect of 5-(4-phenoxybutoxy)psoralen (PAP-1) on glucose uptake in white adipocytes and soleus muscle, and TNFα secretion by white adipocytes from wild-type and ob/ob mice. PAP-1 is a selective inhibitor of Kv1.3, being at least 23-fold selective as an inhibitor of Kv1.3 over other Kv1-family channels and 500-fold selective over Kv2.1, Kv3.1, Kv3.2 and Kv4.2 channels (Schmitz et al., 2005). We report that a concentration of PAP-1 that is not selective for Kv1.3 stimulated glucose uptake and reduced TNFα secretion, but a lower concentration was ineffective, in agreement with the results of others (Beeton et al., 2006; Straub et al., 2011). This raised the question of the target for the non-selective concentration of PAP-1 and led us to investigate the expression of Kv1.1, Kv1.2, Kv1.3 and Kv1.5 in mouse skeletal muscle and white adipose tissue. Kv1.5 is identified as a candidate mediator of the effects of PAP-1 on glucose uptake.

Materials and Methods

Materials

All materials, including PAP-1 (>98% purity), were obtained from Sigma-Aldrich, Poole, UK, unless otherwise stated.

Animals

Housing and procedures were conducted in accordance with the UK Government Animal (Scientific procedures) Act 1986 and approved by the University of Buckingham Ethical review Board. C57Bl/6 and ob/ob mice (Harlan, Bicester, UK), aged 5–6 weeks, were fed standard laboratory chow ad libitum and euthanized 3–4 h after the onset of day light cycle, by a UK Government Animal Scientific Act 1986 schedule 1 method.

RT-PCR

Tissues isolated from wild-type and ob/ob female C57Bl/6 mice were homogenized in Tri-reagent using a ribolyser and total RNA prepared using Qiagen™ minicolumns according to the manufacturer’s instructions. One µg total RNA was reverse-transcribed using avian reverse transcriptase and random priming in a 50 µl reaction. Two µl cDNA was subsequently used per 50 µl PCR reaction as standard. GAPDH was chosen as the housekeeping genes because it showed consistent CT values in adipose tissue, adipocytes and soleus muscle. Gene expression assays were obtained from Applied Biosystems Assay-on-Demand predesigned and optimized assays.

Subsequently, tissues were isolated from wild-type female C57Bl/6 mice. Total RNA was extracted and RT PCR performed by Real Time PCR using optimized Assay-on-Demand (Applied Biosystems) primers and probes for Kv1.1, 1.2, 1.3 and 1.5, relative to an endogenous GAPDH control (Wargent et al., 2013). Expression of each potassium channel was calculated relative to expression of GAPDH in each sample and then the expression levels of Kv1.2 and Kv1.5 were expressed relative to Kv1.3 in the same tissue (Wargent et al., 2013). Multiple Kv channels were not run in the same PCR reaction in order to prevent competition between the cDNA templates for the amplification reactants and to optimize reaction efficiency.

2-Deoxyglucose uptake by soleus muscle

Uptake of 2-deoxyglucose by the soleus muscle was measured by a method that we have described previously (Ngala et al., 2008). Soleus muscles were dissected from both hind legs of wild-type and ob/ob C57Bl/6 male mice. Distal and proximal tendons were tied under resting tension to stainless steel clips and pre-incubated for 60 min in Krebs-Henseleit bicarbonate buffer that contained 10 mM 4-(2-hydroxyethyl)-1-piperazineethanesulfonic acid (HEPES), 5.5 mM glucose and 0.14% (w/v) fatty acid-free bovine serum albumin (BSA) at pH 7.4 and 37 °C. The buffer had been gassed previously with 95% O2: 5% CO2. They were then incubated for 45 min in fresh medium that also contained 2-deoxy[1-14C] glucose (56 mCi mmol-1; 0.1 µCi ml-1) and glucose (5.5 mM), with or without insulin (10 nM for wild-type mice and 100 nM for ob/ob mice) and PAP-1 (3 µM) or a combination of insulin and PAP-1.

At the end of the incubation, the muscles were digested with 1 M NaOH and the digest was neutralised with 1 M HCl. One portion of the mixture was treated with 2.5 volumes of 6% (w/v) perchloric acid to precipitate out proteins, leaving supernatant that contained both 2-deoxyglucose and 2-deoxyglucose-6-phosphate. Another portion was treated with 2.5 volumes of 2.68% (w/v) Ba(OH)2 and 2.51% (w/v) ZnSO4 to precipitate out 2-deoxyglucose-6-phosphate. Radioactivity incorporated into 2-deoxyglucose-6-phosphate was obtained from the difference between that in the two supernatants. 2-Deoxyglucose uptake was calculated by dividing radioactivity in 2-deoxyglucose-6-phosphate by the specific activity of 2-deoxyglucose.

Glucose uptake by adipocytes

Adipocytes were prepared from the parametrial fat pads of wild-type and ob/ob female mice by a method that we have described previously (Zaibi et al., 2010). Tissue was minced and digested with collagenase type II in Krebs-Ringer HEPES buffer containing 10 mM HEPES, 1% bovine serum albumin (fraction V), 2.5 mM CaCl2, 5.5 mM glucose and 200 nM adenosine at pH 7.4 and 37 °C. It was filtered through 250–300 µm nylon mesh. The infranatant was removed and the floating layer of adipocytes was washed four times with a fresh buffer. Adipocytes were concentrated to 40% of final volume of Krebs-Ringer HEPES buffer containing 5% BSA and 0.3 mM glucose and pre-incubated for 45 min under 95% O2: 5% CO2 before dispensing them into 300 µl polyethylene tubes for measurement of glucose uptake.

Glucose transport was measured as described previously (Kashiwagi, Huecksteadt & Foley, 1983). Adipocytes were incubated in Krebs-Ringer HEPES buffer containing BSA, 0.3 mM glucose and D-[U-14C] glucose (0.2 µmol/l; 0.2 µCi/ml), for 1 h at 37 °C in the absence or presence of different concentrations of PAP-1 and insulin. The reaction was stopped by separation of the cells through silicone oil and radioactivity in the cells was measured. Extracellular space was measured in parallel incubations using D-[U-14C] sucrose. Secretion is expressed per mg protein, determined using a method based on the Lowry assay (Bio-Rad, Hemel Hempstead, UK) or per g cells.

TNFα and IL-6 secretion

Adipose tissue from ob/ob and wild-type female mice was minced into pieces weighing about 200 mg and added to 0.3 ml Dulbecco’s modified Eagle’s medium and Ham’s F-12 in a 1:1 mixture (both from Invitrogen, Paisley, UK), supplemented with 0.5% (w/v) endotoxin-free BSA . The incubation medium also contained 1 nM T3 and 5 mM glutamine. The plates were incubated with or without PAP-1 (3 µM) for 30 min. TNFα and IL-6 in the culture medium was assayed using murine ELISA kits (from Diaclone, Besançon, France for TNF; from Invitrogen for IL-6) , following the manufacturer’s instructions. Secretion is expressed per mg protein, determined using the Lowry assay.

Statistical analysis

Glucose uptake was calculated assuming that 2-deoxyglucose and glucose are not distinguished by uptake mechanisms. Data were analyzed by one- or two-way ANOVA followed by Fisher’s least significant difference test, as described in the figure legends, using GraphPad Prism version 5 (GraphPad software, San Diego CA, USA). Results are expressed as means ± SEM.

Results

Glucose uptake in soleus muscle

Insulin (10 nM) and PAP-1 (3 µM) stimulated 2-deoxyglucose uptake in soleus muscle from both wild-type and ob/ob mice (Fig. 1). The combination of insulin and PAP-1 also stimulated uptake compared to baseline in muscle from both wild-type and ob/ob mice (P < 0.001), but it did not have a significantly greater effect than either insulin or PAP-1 alone.

Figure 1 Effect of PAP-1 on glucose uptake in soleus muscles of male (A) wild-type and (B) ob/ob C57Bl6 mice.

Muscles were treated with insulin (wild-type, 10 nM; ob/ob, 100 nM) or PAP-1 (3 µM) alone, or the combination of insulin and PAP-1. n = 8 mice per group for all columns. Data for each genotype were analyzed by one-way ANOVA followed by Fisher’s least significant difference test. ∗ P < 0.05; ∗∗∗ P < 0.001 for effect of PAP-1 compared to controls (Con). †P < 0.05; †††P < 0.001 for effect of insulin (Ins) compared to controls. ‡‡‡P < 0.001 for the effect of the combination of PAP-1 and insulin compared to the absence of both compounds (Con). Note that in both wild-type and ob/ob mice, glucose uptake in the presence of the combination of PAP-1 and insulin was not significantly greater than in the absence of one of the compounds.

Glucose uptake in Adipocytes

PAP-1 (3 µM) stimulated glucose in adipocytes from both wild-type and obese mice in the absence of insulin. In the presence of insulin at a range of concentrations, its effect was significant only in adipocytes from ob/ob mice (Fig. 2). In the absence of PAP-1, insulin stimulated uptake at concentrations of 0.5 nM and higher. The combination of insulin and PAP-1 stimulated uptake compared to baseline, but insulin did not stimulate glucose uptake significantly in the presence of PAP-1. A concentration–response curve conducted using adipocytes from wild-type mice in the absence of insulin showed significant effects of PAP-1 at 1 and 3 µM, but not at lower concentrations (Fig. 3).

Figure 2 Effect of PAP-1 on glucose uptake in adipocytes of female (A) wild-type and (B) ob/ob C57Bl6 mice.

Adipocytes were treated with insulin, PAP-1 (P, 3 µM) alone, or a combination of insulin and PAP-1. n = 8 mice per group for all columns. Data for each genotype were analyzed by two-way ANOVA (sources of variation: PAP-1 and insulin concentrations), followed by Fisher’s least significant difference test, first for the paired columns at the same concentration of insulin, and second to test for an effect of insulin compared to the value obtained in the absence of insulin but in the presence of the same concentration (0 or 3 microM) of PAP-1. ∗ P < 0.05; ∗∗P < 0.01 for effect of PAP-1 compared to controls at the same concentration of insulin (C). †P < 0.05; ††P < 0.01; †††P < 0.001 for effect of insulin compared to no insulin at the same concentration of PAP-1.

Figure 3 Concentration-response curve for the effect of PAP-1 on glucose uptake in the absence of insulin in adipocytes of female wild-type C57Bl6 mice.

n = 8 mice per group for all columns. Data were analyzed by one-way ANOVA followed by Fisher’s least significant difference test against the control value. ∗P < 0.05; ∗∗∗P < 0.001 for effect of PAP-1 compared to baseline.

TNFα and IL-6 secretion

PAP-1 (3 µM) suppressed the secretion of TNFα by 50% in adipocytes from both wild-type and ob/ob mice (Fig. 4). PAP-1 did not affect IL-6 secretion (results not shown). Surprisingly, secretion of TNFα from ob/ob adipocytes was lower than from wild-type adipocytes. Others have found a higher concentration of TNFα in adipocytes from ob/ob compared to wild-type adipocytes (Galvez, San Martin & Rodriguez, 2009).

Figure 4 Effect of PAP-1 (3 µM) on TNFα secretion by adipocytes from female wild-type and ob/ob mice.

n = 6 mice per group for all columns. Data were log10-transformed so that variances were not significantly different between groups prior to analysis by two-way ANOVA (sources of variation: genotype and PAP-1 concentration), followed by Fisher’s least significant difference test. ∗P < 0.05; ∗∗P < 0.01 for effect of PAP-1 compared to control (Con).

Expression of Kv1mRNA

Using reverse transcription PCR, Kv1.3 mRNA was detected in liver, gastrocnemius muscle, soleus muscle and parametrial white adipose tissue of female wild-type and ob/ob mice, except that expression was not significantly above baseline in gastrocnemius muscle from wild-type mice (Fig. 5A). Expression was higher in liver, gastrocnemius muscle and soleus muscle, but not adipose tissue, of ob/ob mice compared to wild-type mice. Kv1.5 mRNA was detected in liver, gastrocnemius muscle, soleus muscle and parametrial white adipose tissue of female wild-type and ob/ob mice, except that expression could not be detected in livers from wild-type mice (Fig. 5B). Expression was higher in liver and soleus muscle of ob/ob mice compared to wild-type mice.

Figure 5 Kv1.3 and Kv1.5 mRNA levels in liver, gastrocnemius, soleus and white adipose tissue of female (A) wild-type and (B) ob/ob mice.

n = 3 mice for all columns. Expression of each potassium channel was calculated relative to expression of GAPDH in each sample. Data were log10-transformed prior to statistical analysis because the data obtained for Fig. 6 suggested that widely distributed data did not follow a normal distribution without this transformation. Analysis was by two-way ANOVA (sources of variation: genotype and tissue) for each gene, followed by Fisher’s least significant difference test for each pair of wild-type and ob/ob values. Unpaired t-tests conducted on untransformed data gave the same significant differences. ∗P < 0.05; ∗∗P < 0.01; ∗∗∗P < 0.001 compared to wild-type mice.

After it was found that PAP-1 did not stimulate glucose uptake by adipocytes at concentrations that should be sufficient to activate Kv1.3, Real-Time PCR was used to detect mRNA for Kv1.1, Kv1.2, Kv1.3 and Kv1.5 in each of adipose tissue, adipocytes and soleus muscle. Assuming PCR reaction efficiencies of 1.0 for each gene, Kv1.5 was 30-, 15- and 78-fold more highly expressed than Kv1.3 in adipose tissue, adipocytes and soleus muscle respectively. Kv1.2 mRNA could not be detected in adipocytes and appeared to have lower expression than Kv1.3 in adipose tissue (Fig. 6). Kv1.1 could not be detected in adipose tissue, adipocytes or soleus muscle.

Figure 6 Relative expression of Kv1.2, Kv1.3 and Kv1.5 expression in (A) adipose tissue, (B) adipocytes and (C) soleus muscle of wild-type mice.

Relative expression is given assuming that PCR reaction efficiencies were 1.0 for each gene. The expression of each Kv channel was expressed relative to GAPDH in the same sample and then relative to Kv1.3. ΔCT GAPDH–ΔCT Kv1.3 was 4.43 ± 0.31 for adipose tissue, 6.32 ± 0.34 for adipocytes and 9.8 ± 0.70 for soleus muscle (n = 6). Data were log10-transformed prior to statistical analysis because variances were clearly higher for Kv1.5, which was far more highly expressed than Kv1.2 or Kv1.3. Thus the Bartlett and F-tests showed significant differences in variance for the adipose tissue and adipocyte data respectively when these data were not transformed. Analysis was by two-way ANOVA (sources of variation: gene and tissue), followed by Fisher’s least significant difference test for adipose tissue and soleus muscle. Kv1.2 was undetectable in adipocytes, so Kv1.3 and Kv1.5 were compared by unpaired t-test. n = 4 mice for soleus muscle and 6 mice for adipose tissue and adipocytes. ∗∗P < 0.01; ∗∗∗P < 0.001.

Discussion

A variety of evidence suggests that inhibition of Kv1.3 potassium channels might be effective in the treatment of insulin resistance, and thereby of type 2 diabetes (Choi & Hahn, 2010; Upadhyay et al., 2013). This includes the finding that margatoxin, which inhibits members of the Kv1 family, including Kv1.3, lowered blood glucose in wild-type but not Kv1.3−/− mice (Xu et al., 2004). The authors linked this finding to the demonstration that margatoxin inhibited the secretion of TNFα and IL-6 by isolated adipocytes in adipose tissue and skeletal muscle after it was administered to wild-type but not Kv1.3−/− mice. The tissues that contain the Kv1.3 channels at which the inhibitors exert their initial effects have not been well-defined, however.

We investigated whether the initial effects of Kv1.3 inhibitors might be on Kv1.3 channels in adipocytes or skeletal muscle using PAP-1. PAP-1 inhibits Kv1.3 with 33-fold selectivity over Kv1.1, 125-fold selectivity over Kv1.2 and 23-fold selectivity over Kv1.5 (Schmitz et al., 2005). Inhibition of K channels might be expected to lead to cell depolarization and an increased demand for energy. We found that PAP-1 at the high concentration of 3 µM stimulated glucose uptake in the absence of insulin in isolated soleus muscle and adipocytes from both wild-type and ob/ob mice. PAP-1 also stimulated glucose uptake in the presence of a range of concentrations of insulin in adipocytes from ob/ob mice. The same concentration of PAP-1 (3 µM) also reduced the secretion of TNFα by adipocytes, potentially consistent with the suggestion of others (Xu et al., 2004) that this is a mechanism by which blockade of Kv1.3 improves insulin sensitivity. However, when we investigated the concentration–response curve for the effect of PAP-1 on glucose uptake, we found that, whilst 1 µM was effective, a concentration of 300 nM, which is still 150-fold higher than its EC50 value for blockade of murine Kv1.3 (Schmitz et al., 2005), had no effect on glucose uptake.

These experiments were conducted using tissues from female mice, except that soleus muscles from male mice were used for studying glucose uptake. These choices were in line with our previous work, allowing more accurate quality control by comparison with our historic data. Most studies of the effects of Kv1.3 inhibitors on metabolism have been conducted using male mice, but two studies have shown that deletion of the Kv1.3 gene has similar metabolic effects in male and female mice (Xu et al., 2003; Tucker, Overton & Fadool, 2008).

Our results are consistent with a report that PAP-1 failed to lower blood glucose or enhance the glucose lowering effect of insulin when administered as a single dose or over five days to ob/ob or db/db mice in amounts that gave blood levels that would selectively block Kv1.3. Moreover, 10 nM PAP-1, a concentration fivefold higher than its IC50 value for inhibition of Kv1.3 channels, did not alter K+ currents in a human skeletal muscle cell line or glucose uptake in mouse 3T3-L1 adipocytes (Straub et al., 2011). Similarly, margatoxin did not stimulate glucose utilization by rat epididymal or mesenteric adipocytes (Beeton et al., 2006) when used at concentrations (1 or 100 nM) well above its EC50 value (30 pM) for inhibition of Kv1.3 (Garcia-Calvo et al., 1993). Upadhyay et al. (2013) have reported that the highly selective Kv1.3 inhibitor ShK-186 reduces obesity and hyperglycemia in mice fed on an obesogenic diet, but they too rule out an involvement of white adipocyte or skeletal muscle Kv1.3 channels, mainly because ShK-186 did not stimulate glucose uptake by white adipose tissue or skeletal muscle. ShK-186 doubled glucose uptake in brown adipose tissue, a site of high glucose uptake and uncoupled oxidative phosphorylation, making this a more likely explanation for the anti-obesity and anti-hyperglycemic effects of ShK-186, in their view. The same group had previously reported that ShK-186 also failed to stimulate glucose uptake by rat white adipocytes (Beeton et al., 2006).

Although we do not believe that PAP-1 affected glucose uptake by inhibiting Kv1.3, we did detect Kv1.3 mRNA in murine adipose tissue and skeletal muscle (and also liver). Others appear to disagree as to whether Kv1.3 protein is expressed in murine adipose tissue, but this may be because one group used isolated epididymal adipocytes (Li et al., 2006) and the other visceral adipose tissue (Upadhyay et al., 2013). No Kv1.3 protein was detected in murine skeletal muscle (Upadhyay et al., 2013) or in samples of adipose tissue and skeletal muscle from two normal and two type 2 diabetic human subjects (Straub et al., 2011). One possible interpretation of these reports and our findings is that the expression of Kv1.3 is very low in adipose tissue and skeletal muscle, and is sometimes undetectable at the protein level, despite being detectable at the mRNA level. We found more Kv1.3 mRNA in tissues from ob/ob than wild-type mice, except for white adipose tissue, but given the relative amounts of adipose tissue in the wild-type and obese animals, the total amount of Kv1.3 mRNA in adipose tissue per animal may be higher in the ob/ob mice. We also found more Kv1.5 mRNA in liver and soleus muscle of ob/ob compared to lean mice. These results would be consistent with expression of either channel contributing to insulin resistance in ob/ob mice.

Levels of Kv1.5 mRNA appeared much higher than those of Kv1.3 in both adipocytes and soleus muscle. It is well-established that Kv1.5 is expressed in skeletal muscle (Kang et al., 2009) but its expression in adipose tissue or adipocytes has not been reported. We cannot exclude the possibility that our results are influenced to some extent by the PCR reaction efficiency differing between genes, but this seems unlikely to alter our conclusion that the expression of Kv1.5 was much higher than that of Kv1.3 or Kv1.2 (Irwin et al., 2012). In any event, irrespective of the precise differences in expression of the Kv genes, our findings raise the possibility that Kv1.5 was the channel that PAP-1 inhibited to stimulate glucose uptake and inhibit TNFα secretion. If so, Kv1.5 inhibitors might have potential in the treatment of type 2 diabetes. PAP-1 has been reported to inhibit Kv1.5 with an EC50 value of 45 nM (Schmitz et al., 2005), but it failed to stimulate glucose uptake in adipocytes when present at a concentration of 300 nM. The 6.7-fold lower EC50 value of 45 nM was, however, measured with Kv1.5 from a different species and under very different conditions: it was obtained by applying depolarizing pulses to mouse erythroleukaemia (MEL) cells stably transfected with human Kv1.5 and measuring the reduction in area under the current curve (Grissmer et al., 1994). By contrast, no depolarizing stimulus was applied in our work and we used mouse tissue. Whilst the EC50 value of 2 nM for PAP-1 at Kv1.3 was also determined by an electrophysiological technique, it did use murine Kv1.3 (Schmitz et al., 2005), and it is less likely that the much higher (150-fold) discrepancy between this value and the minimum effective dose of PAP-1 for stimulation of glucose uptake can be explained in terms of the different test systems. PAP-1 is less potent at Kv1.2 (EC50 = 250 nM) than Kv1.5 (Schmitz et al., 2005), and Kv1.2 mRNA could not be detected in adipocytes using Real Time RCR, ruling it out as a candidate target for PAP-1 in this tissue. We cannot rule out the possibility that Kv1.4, Kv1.6 or Kv1.7 could be the targets of PAP-1 in adipocytes and skeletal muscle, but Kv1.4 is poorly expressed in human adipocytes (You et al., 2013), and 100 nM margatoxin did not stimulate glucose uptake in rat epididymal adipocytes (Beeton et al., 2006), despite having an EC50 value for inhibition of Kv1.5 of 5 nM (Garcia-Calvo et al., 1993).

We therefore conclude that although Kv1.3 mRNA could be detected in murine white adipose tissue, white adipocytes and red skeletal muscle, inhibition of Kv1.3 does not directly affect glucose uptake in white adipocytes or skeletal muscle. Blockade of Kv1.5 channels in white adipocytes or skeletal muscle might, however, stimulate glucose uptake. Studies using novel Kv1.5 inhibitors might be informative, but their selectivities have been given only relative to other cardiac K channels and not to other Kv1 channels (Ford et al., 2013; Pavri et al., 2012). We do not exclude the possibility that compounds might be useful in the treatment of obesity and insulin resistance, by inhibiting Kv1.3 in other tissues.

The authors are grateful to Dr Claire Cornick for assisting with the experiments, to Professor John Clapham for discussion of the manuscript and to Mrs Anita Roberts for help with the husbandry of the animals.

Additional Information and Declarations

Competing Interests

Author Contributions

Animal Ethics

The authors declare there are no competing interests.

Robert A. Ngala, Mohamed S. Zaibi and Claire J. Stocker conceived and designed the experiments, performed the experiments, analyzed the data, contributed reagents/materials/analysis tools, wrote the paper, prepared figures and/or tables, reviewed drafts of the paper.

Kenneth Langlands conceived and designed the experiments, performed the experiments, analyzed the data, contributed reagents/materials/analysis tools, prepared figures and/or tables, reviewed drafts of the paper.

Jonathan R.S. Arch conceived and designed the experiments, analyzed the data, wrote the paper, prepared figures and/or tables, reviewed drafts of the paper.

Michael A. Cawthorne conceived and designed the experiments, reviewed drafts of the paper, manuscript revision and approval.

The following information was supplied relating to ethical approvals (i.e., approving body and any reference numbers):

University of Buckingham Ethical Review Board. Animals were killed by an approved method to obtain tissues. Ethical approval numbers for housing and obtaining of tissues: 05/045; 06/016; 06/017; 14/010. Project licence PPL 70/7189.

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
