# Peer review of "Stimulation of glucose uptake in murine soleus muscle and adipocytes by 5-(4-phenoxybutoxy)psoralen (PAP-1) may be mediated by Kv1.5 rather than Kv1.3"

_PeerJ, doi:10.7717/peerj.614_

## Round 0.1 · original submission · Major Revisions

We deeply apologize for the delay in receiving complete reviews. All of the reviewers commented that it would be important to evaluate the roles or at least expression of the other Kv1.x channels in these tissues. Loss of function approaches (e.g. siRNA or a knockout) would be helpful, if feasible. Because the mRNA data are possibly discrepant with previously reported data regarding protein expression of Kv1.3 in metabolic tissues, the reviewers request additional data regarding Kv1.3 protein expression in muscle, adipose and liver. The reviewers also request clarifications to the methodology and sources of materials as well as alternate statistical tests for some experiments.

Please respond to all of the reviewer comments in your rebuttal letter with additional data as appropriate.

Reviewer 1 ·

Basic reporting

Article meets all these aspects with the exception of the figure legends. I have noted in the comments to the authors how this should be modified.

Experimental design

Article does not meet several of these aspects. It is not clear how high pharmacological doses of this drug change the uptake of glucose if the target of action is left unidentified. This is a major drawback of the research as fully described in the comments to the authors. There are several reagents (the main drug PAP-1 itself) for which the supplier is not clear. There are no ethical concerns.

Validity of the findings

Article does not meet several of these aspects. There are difficulties in deciphering which statistical metric was applied and other data sets in which the wrong metric was selected. Speculation as to the target for PAP-1 in these studies is made without evidence.

If inconclusive results are acceptable, then this report definitely falls into this category because it is not know how PAP-1 is causing a change in glucose uptake or what the role of Kv1.3 mRNA would be as found in tissues that have previously reported the absence of this channel at the protein level.

Comments for the author

The Kv1 inhibitor 5-(4-phenoxybutoxy)psoralen (PAP-1) stimulates glucose uptake in soleus muscle and adipocytes of wild-type and obese mice

Recent literature has demonstrated that inhibition of Kv1.3 channel, genetically and pharmacologically, can reduce body weight, adiposity, TNF alpha expression, and insulin resistance among other metabolic metrics. This suggests that drugs designed to target this ion channel may be very useful therapeutically to treated obesity and type II diabetes. In this report, the authors add to this knowledge by testing a selective inhibitor of Kv1.3 (PAP-1) in wild-type and ob/ob mice and discover that it increases glucose uptake in muscle and adipocytes, but at concentrations that are not pharmacologically specific to this subtype of Shaker channel. Another report (Staub et al., 2011) has already demonstrated that application of PAP-1 at concentrations known to effectively block Kv1.3 are NOT effective in increasing glucose uptake peripherally but do modify uptake centrally. While the report discovers that high concentrations of this inhibitor DO modify glucose uptake (and lower concentrations do not), the target for this action is completely unknown, which leaves the work really hanging.

Major Comments:

1. The identity of the target for 1-3 uM PAP-1 stimulation to increase glucose uptake in these two different tissues is really unknown. The authors state that is more likely due to inhibition of Kv1.1, Kv1.2, or Kv1.5 (other related Shaker family members) without ANY experimental evidence. A better design would be to test for glucose uptake in gene-targeted deleted models to demonstrate that the small peptide molecule is not targeting that particular channel. For example, if glucose uptake was increased in Kv1.3-/- mice, then there would be strong evidence that these high concentrations of peptide were working through a different target.
2. PAP-1 blocks by a use-dependent mechanism from a C-type inactivated state whereas the comparisons that you are making with ShK-186 and MgTx are not well aligned. These later toxins act by blocking the vestibule of the P region using a helix/kink/helix fold that is unique to K channel members. Because you are not activating your tissue preparations, perhaps the well characterized block by PAP-1 is different than what you anticipate by just incubation with the drug. Another complication is that these authors are using an in vitro preparation whereas the comparative studies are largely in vivo.
3. It is a concern that the authors report Kv1.3 mRNA in adipose tissue, skeletal, and liver whereas others do not report Kv1.3 protein in these tissues. The authors need to better justify these discrepancies with potential explanation and impact.
4. There is a problem with the statistical metrics applied to each of the data across the four figures. First, it is not known what statistical metric is applied as the methods describe a choice between either a 1-way ANOVA or a t-test. Second, the sample size of the first figure would only allow a dF of 2, thus it really needs to be non-parametric. This data set also uses two variables (tissue type and type of mouse) so it needs to be a 2-way design. For the data in Figures 2 and 3, it would be stronger to have a comparison of a single 2-way ANOVA (type of mouse and drug treatment) than two individual 1-way ANOVA, each divided out by type of mouse. In Figure 3, even if you keep the analysis as it is (divided by type of mouse), then it still has to be a 2-WAY ANOVA because you have 2 treatments (control/pap and dose of insulin). The Figure 4 looks as if it were done properly (if a 1-way ANOVA were applied; treatment being drug concentration) but the metric is not specified so it is not certain. For the last figure (5), there are two treatments again (type of mouse and drug treatment) thus a 2-WAY is required.

Minor Comments:

1. PAP-1 is a central reagent of this study. The authors need to specify source (is it Sigma), lot, and purity grade.
2. Abbreviations need to be spelled out at first use: BSA, HEPES, KRH, SEM
3. Why do the skeletal muscle vs. adipocyte buffers use different concentrations of glucose to examine glucose uptake (5.5 vs. 0.3 mM) - just curious. Also the plus insulin conditions across these tissues is also differential (10 nM vs. 100 nM) - imagine it is attributed to resting or fasted state of glucose/insulin in these tissues.
4. There are a number of reagents that are likely not coming from Sigma Chemical that need to be specified as to distributor source. For example, Dulbecco’s MEM, Ham’s F-12, BSA, T3, glutamine, ELISA kits, etc.
5. The type of statistical metric applied to each experiment needs to be clarified instead of clustered in a paragraph that states that either a 1-way ANOVA or a unpaired t-test was applied to all data. This also needs to be clarified in the figure legends.
6. Typo, page 8, line 119 - KV1.3 needs to be Kv1.3 lower case.
7. Typo, page 17, last line of legend 2 - in the in the
8. Shaker is typically italicized as gene name.

Reviewer 2 ·

Basic reporting

The paper by Ngala et al. describes the effect of specific Kv1.3 inhibitor PAP-1 on glucose uptake in wt and obese mice. In particular they address the effect of this substance on soleus muscle and adipocytes.
The topic of the paper is interesting, however the paper is not written in a clear way and several important control experiments are missing. The paper is not acceptable in its present form.

Experimental design

1) The main conclusion of the paper states that “stimulation of glucose uptake is more likely due to inhibition of Kv1.1, Kv1.2 or Kv1.5 than to inhibition of Kv1.3 channels”. However, it is not mentioned and/or investigated whether Kv1.1, Kv1.2 and Kv1.5 are expressed in the studied tissues. This should be done using RT-PCR of Western blot.
2) In order to demonstrate that the effect of PAP-1 at the relatively high concentration used is correlated to its inhibition of Kv channels, Kv1.3 and possibly the other Kv channels should be silenced by siRNA (commercially available).
3) The highly selective and potent Kv1.3 inhibitor ShK-186 should be used at least in some experiments to clarify /confirm the role of Kv1.3.
4) By detecting Kv1.3 mRNA in murine adipose tissue, skeletal muscle and liver, the authors propose the presence of this channel in these tissues. This important result should be confirmed by Western blot analysis as well.

Validity of the findings

1) Results are described only briefly, rationale behind the experiments is not explained, nor interpretation of the results is given.
2) Figure legends do not give sufficient details to understand the data shown. For example in figures “lean mice” are reported, but in the main text there if no reference to any lean mice. Another example: for how long time was PAP-1 applied in the glucose uptake experiments?

Comments for the author

1) The paper should be rewritten clearly stating the objectives of the work and the rationale why these experiments were carried out. Superfluous information could be deleted. E.g. it is mentioned in the introduction that Kv1.3 is phosphorylated, but this information is not essential in order to understand the paper. Viceversa, it should be better described how secretion of interleukin-6 (IL-6) and tumour necrosis factor α (TNFα) from adipose tissue ( Xu et al . , 2004 ) are thought to influence the rapid effect of Margatoxin.
2) Please re-write the sentence” This might be due in part to their also being protected from diet-induced obesity, despite eating no less than wild-type mice ( Xu et al . , 2004 )”.
3) The authors write: “The authors link these findings to activation of brown adipose tissue ( Upadhyay et al . , 2013 ).” Please specify what does this mean. Does it refer to activation of metabolism in brown adipose tissue?
4) Please unify writing of the channel: sometimes it is written as Kv1.3 (correct) sometimes KV1.3.
5) Please explain why reduced JNK activity leads to reduced blood glucose level.
6) The authors conclude: “Our results in no way preclude the possibility put forward by others ( Upadhyay et al . , 2013 ) that selective inhibition of Kv1.3 channels reduces TNFα secretion by T cells and macrophages in adipose tissue, thereby reducing adipose tissue inflammation and improving insulin sensitivity.” Is this statement compatible with the fact that the authors observe changes in TNFalfa secretion after 30 minutes of treatment?

·

Basic reporting

The article is well written and the scientific question laid out clearly.

However, to make the article a full "unit of publication" it would really support the authors findings, namely that the effects of PAP-1 on glucose uptake and TNF-alpha are not due to Kv1.3 blockade but due to blockade of other Kv1 family channels, if the authors could provide some evidence for Kv1.1, Kv1.2 and Kv1.5 expression in mouse adipocytes. Without this evidence it is really hard to judge for the reader whether the "significant" Kv1.3 mRNA expression in adipocytes is really significant or just "noise". Why not analyze all Kv1 channels at the same time and compare expression levels? Of course the ideal techniques would be IHC and electrophysiology, but as a reviewer I recognize that these techniques are very difficult to apply to adipocytes.

Experimental design

The experiments are well designed and the methods described with sufficient detail.

However, as mention above the authors should really compare expression levels of Kv1.3, K1.1, Kv1.2 and Kv1.5 to support their conclusion that PAP-1's effects on adipocytes are due to inhibition of other Kv1 channels.

Validity of the findings

See above. It is currently really not clear who significant the Kv1.3 expression is.

Comments for the author

No additional comments.

---

## Round 0.2 · Minor Revisions

We apologize for the delay in responding with a decision.

Overall the manuscript is clearly written and the description and discussion of the results appears consistent with the data.

Two minor issues require clarification or revision with different methodology.

1. qPCR data are provided in arbitrary units (Fig 5-6) for Kv1.2, Kv1.3, Kv1.5. The methods reference use of gene specific primers/ probes normalized to Gapdh for each target gene. This is perfectly valid for comparisons of individual Kv isoform expression among tissues (Fig 5), but as written does not appear valid for comparison of multiple targets to each other (Fig 6). For example, one cannot rule out that differential PCR efficiency resulted in the evident enrichment of Kv1.5 relative to Kv1.3 in adipose. To substantiate this claim, an absolute quantification method is required that would allow one to estimate actual copy #, either by qPCR using appropriate cDNA standards for each gene or perhaps by using radiolabelled probes again with standards to demonstrate similar probe binding efficiency. As this is a major claim of the paper, it should be strengthened with more quantitative data if analysis of protein abundance is not possible.

2. Why were female animals used for adipocyte harvest but males for skeletal muscle? Please comment in the methods.

As a side note, I'd suggest using the term "euthanized" rather than "killed" in the animal methods section.

---

## Round 0.3 · accepted · Accept

Thank you for your revision and for contributing to PeerJ.